# One-Year Outcomes after Myval Implantation in Patients with Bicuspid Aortic Valve Stenosis—A Multicentre Real-World Experience

**DOI:** 10.3390/jcm12062398

**Published:** 2023-03-20

**Authors:** Ahmed Elkoumy, John Jose, Christian Juhl Terkelsen, Henrik Nissen, Sengottuvelu Gunasekaran, Mahmoud Abdelshafy, Ashok Seth, Hesham Elzomor, Sreenivas Kumar, Francesco Bedogni, Alfonso Ielasi, Shahram Arsang-Jang, Santosh Kumar Dora, Sharad Chandra, Keyur Parikh, Daniel Unic, Andreas Baumbach, Patrick Serruys, Osama Soliman

**Affiliations:** 1Health Service Executive and CORRIB Core Lab, Discipline of Cardiology, Saolta Group, Galway University Hospital, University of Galway, H91 V4AY Galway, Ireland; ahmed.elkoumy@universityofgalway.ie (A.E.);; 2Islamic Center of Cardiology, Al-Azhar University, Cairo 11651, Egypt; 3Christian Medical College & Hospital, Vellore 632004, India; 4Department of Cardiology, Aarhus University Hospital, 8200 Aarhus, Denmark; 5Department of Cardiology, Odense University Hospital, 5000 Odense, Denmark; 6Department of Cardiology, Apollo Main Hospital, Greams Road, Chennai 600006, India; 7Department of Cardiology, Al-Azhar University, Cairo 11311, Egypt; 8Fortis Escorts Heart Institute, New Delhi 110025, India; 9Department of Cardiology, Apollo Hospitals, Apollo Health City, Jubilee Hills, Hyderabad 500050, India; 10Department of Cardiology, IRCCS Policlinico San Donato, 20097 Milan, Italy; 11Clinical and Interventional Cardiology Unit, Istituto Clinico Sant’Ambrogio, 20149 Milan, Italy; 12CÚRAM—SFI Research Centre for Medical Devices, H91 TK33 Galway, Ireland; 13Asian Heart Institute, Mumbai 400051, India; 14Department of Cardiology, King George’s Medical University, Lucknow 226003, India; 15Care Institute of Medical Sciences, Ahmedabad 380060, India; 16Department of Cardiac and Transplant Surgery, University Hospital Dubrava, 10000 Zagreb, Croatia; 17Barts Heart Centre, William Harvey Research Institute, Queen Mary University of London, London EC1M 6BQ, UK; 18National Heart and Lung Institute (NHLI), Imperial College London, London SW7 2AZ, UK

**Keywords:** aortic stenosis, bicuspid aortic valve, BAV, transcatheter aortic valve implantation, Myval

## Abstract

Background: Bicuspid aortic valve (BAV) affects approximately 1.5% of the general population and is seen in nearly 50% of candidates for aortic valve replacement (AVR). Despite increasingly utilised transcatheter aortic valve implantation (TAVI) in aortic stenosis (AS) patients, its use among patients with severe bicuspid AS is limited as BAV is a heterogeneous disease associated with multiple and complex anatomical challenges. Aim: To investigate the one-year outcomes of TAVI using the balloon-expandable Myval transcatheter heart valve (THV) (Meril Life Sciences Pvt. Ltd., Vapi, India) in patients with severe bicuspid AS. Methods and results: We collected data from consecutive patients with bicuspid AS who underwent TAVI with the Myval THV and had at least one-year follow-up. Baseline characteristics, procedural, and 30-day echocardiographic and clinical outcomes were collected. Sixty-two patients were included in the study. The median age was 72 [66.3, 77.0] years, 45 (72.6%) were males, and the mean STS PROM score was 3.2 ± 2.2%. All TAVI procedures were performed via the transfemoral route. The median follow-up duration was 13.5 [12.2, 18.3] months; all-cause mortality was reported in 7 (11.3%) patients and cardiovascular hospitalisation in 6 (10.6%) patients. All-stroke was reported in 2 (3.2%), permanent pacemaker implantation 5 (8.3%), and myocardial infarction 1 (1.6%) patients. The echocardiographic assessment revealed a mean pressure gradient of 10 [8, 16.5] mmHg, effective orifice area 1.7 [1.4, 1.9] cm^2^, moderate AR in 1 (2%), mild AR in 14 (27%), and none/trace AR in 37 (71%). In total, 1 patient was diagnosed with valve thrombosis (2.1%), Stage II (moderate) haemodynamic deterioration was seen in 3 (6.4%), and stage III (severe) haemodynamic deterioration in 1 (2.1%) patient. Conclusions: TAVI with the Myval THV in selected BAV anatomy is associated with favourable one-year hemodynamic and clinical outcomes.

## 1. Introduction

Transcatheter aortic valve implantation (TAVI) has become an established intervention strategy for patients with severe aortic stenosis (AS) [1,2]. Extension of TAVI into a wider spectrum of surgical risk will include a younger patients group in whom bicuspid aortic valve (BAV) is more common [3,4]. Based on multiple surgical aortic valve replacement (SAVR) studies, approximately 50% of patients with aortic valve (AV) dysfunction and indicated for aortic valve replacement (AVR) have BAV anatomy [5].

Neither the recent 2020 ACC/AHA nor 2021 ESC/EACTS guidelines have included patients with BAV in their specific recommendations. Furthermore, both guidelines considered BAV as unfavourable anatomy for TAVI [1,2].

BAV shows a wide spectrum of heterogenicity regarding the anatomy, which is one of the main constraints facing the inclusion of BAV in randomised trials [6]. The BAV heterogenicity may be due to the association of eccentric valve opening, the fused raphe position (Sievers’ classification: type 0, type-1, and type-2), sinus of Valsalva asymmetry, aortic dilatation, and extensive calcification at multiple levels (leaflet, raphe, annulus, and LVOT) [7].

Data about the feasibility of TAVI in BAV are obtained mainly from several observational cohort studies and registries among selected patients with different surgical risk groups and using various transcatheter heart valves (THV) [8,9,10,11,12,13,14].

To date, BAV patients have not been included in a randomised trial dedicated to TAVI either to test the safety in comparison to surgical aortic valve replacement (SAVR), comparison to tricuspid AV (TAV), or to compare the safety and efficacy between different TAVI devices [3]. Furthermore, there is no consensus on the best design for the required trials [15].

The reports of mid and long-term outcomes of TAVI in BAV are scarce. The importance of such reports is to present the main long-term outcomes in terms of freedom from valve failure-related mortality, aortic valve re-intervention, and freedom from severe hemodynamic deterioration, in addition to freedom from stroke and significant bleeding secondary to the use of antithrombotic therapy [16]. In a recent paper by Elkoumy et al. [10], the 30-day safety and performance of the balloon-expandable (BE) Myval THV (Meril Life Sciences Pvt. Ltd., Vapi, India) according to the updated valve academic research consortium-3 (VARC3) in patients with severe bicuspid AS was reported. The aim of this paper is to present the one-year clinical and hemodynamic outcomes of the previously published cohort of patients with BAV treated with Myval THV [10]. (Ref).

## 2. Patients and Methods

Data of patients with BAV treated with TAVI using the BE Myval THV and included in the first 30-day safety and efficacy report [10] were collected retrospectively from 10 centres in India (*n* = 5), Denmark (*n* = 2), Italy (*n* = 2), and Croatia (*n* = 1). A list of participating centres, collaborators, and numbers of patients included is mentioned previously [10]. The details of the included cohort, data collection, and procedural details were previously described [10]. Clinical and hemodynamic outcomes, at one-year follow-up, including site-reported echocardiographic assessment, were collected and reassessed by the CORRIB Core Laboratory (University of Galway, Galway, Ireland). Patient follow-up was performed according to each site’s local protocol (in-person visits or structured telephone calls). This registry was conducted as an academic collaboration among the participating centres with the aim of reporting the follow-up of the previously reported cohort (previously mentioned) [10].

### 2.1. Procedural Characteristics

Procedural characteristics and 30-day outcomes were described previously [10].

### 2.2. Endpoints and Definitions

Clinical endpoints were collected and reported according to the updated VARC-3 [16], including all-cause mortality, all-stroke, and hospitalisation.

The left ventricular (LV) systolic function and device hemodynamic assessments, including the severity of paravalvular aortic regurgitation (AR), were assessed and graded using transthoracic echocardiography (TTE) according to established guidelines [16,17,18,19,20]. Aortic regurgitation was reported as total AR and was classified as none/trace, mild, moderate, or severe. Haemodynamic valve deterioration was assessed in comparison to the 30-day reported haemodynamic data.

### 2.3. Statistical Analysis

Continuous variables were presented as mean and standard deviation (SD), or median and interquartile range (IQR), after testing the data distribution using the Shapiro– Wilk normality test. Categorical variables were presented as frequencies and percentages. Overall survival was measured from the time of the index TAVI procedure to the time of death or the last follow-up. Analysis was performed using the IBM^®^ SPSS^®^ Statistics version 27 (IBM Corp. in Armonk, NY, USA).

## 3. Results

### 3.1. Study Population and Baseline Characteristics

The study comprised 62 patients with severe bicuspid AS who underwent TAVI using the BE Myval THV system. The median [IQR] of age was 72 [66.3, 77.0], and most were male (72.6%). Among the 62 patients, 80.6% had type 1-a BAV, 16.1% type 0, and 3.2% type 2. The median Society of Thoracic Surgeons (STS) Predicted Risk of Mortality score was 3.2 ± 2.2%. Baseline patient characteristics are summarised in Table 1.

### 3.2. Clinical Outcomes

The median follow-up duration was 13.5 [12.2, 18.3] months from the index TAVI procedure. All-cause mortality until the last follow-up was reported in 7 (11.3%) patients. Cardiovascular mortality, which is not related to the THV or TAVI procedure, occurred in 7 (4.8%) and non-cardiovascular mortality in 4 (6.5%) patients.

Rehospitalisation related to the TAVI procedure occurred in 1 (1.8%) patient, and other cardiovascular rehospitalisation was reported in 8.8% of patients (Table 2). All-stroke was reported in 2 (3.2%) patients, and both were ischemic strokes.

Myocardial infarction was reported in 1 (1.6%) patient. A permanent pacemaker was implanted in 5 (8.3%) patients. Major bleeding was reported in 1 patient (1.6%), but no transfusion was required. At the time of the last follow-up, three patients were symptomatic with NYHA class III. Among the included population, five patients were diagnosed with SARS-COVID-2 infection, and only one patient died due to COVID-2 infection. No endocarditis, no acute kidney injury (AKI), or THV reintervention until the last follow-up visit.

### 3.3. Echocardiographic Outcome

At the last echocardiographic follow-up visits, the haemodynamic data were median transprosthetic maximum velocity (Vmax) of 2.1 [1.6, 2.6] m/s, mean pressure gradient (mPG) of 10 [18, 16.5] mmHg, and effective orifice area (EOA) of 1.7 [1.4, 1.9] cm^2^ (Table 3), (Figure 1). The total aortic regurgitation (AR) in post-TAVI TTE assessment was non/trace AR in 71%, mild AR in 27%, moderate in 2%, and there were no cases of severe AR (Table 3), (Figure 1).

### 3.4. Bioprosthetic Valve Deterioration

Assessment of bioprosthetic valve deterioration was defined according to VARC-3 definitions and categories [16].

Stage I, with morphological valve deterioration in the form of clinical valve thrombosis, was diagnosed by multidetector computed tomography (MDCT) scan in 1 (2.1%) patient.Stage II haemodynamic valve deterioration was detected in 3 (6.4%) patients.Stage III haemodynamic valve deterioration was detected in 1 (2.1%) patient.

## 4. Discussion

To the best of our knowledge, this is the first report of the one-year follow-up (up to 3 years) of patients with severe bicuspid AS treated with TAVI using the BE Myval THV.

The main results of the current report are: (1) all-cause mortality was reported in 7 (11.3%) patients, (2) all-cause rehospitalisation was reported in 12 (21.1%), while procedure-related hospitalisation was only seen in 1 (1.8%) patient, (3) all-stroke was 3.2%, (4) permanent pacemaker was implanted in 8.3%, and (5) haemodynamic outcomes showed mPG of 10 mmHg, EOA of 1.7 cm^2^, and moderate AR in 2%.

The data about the mid and long-term outcomes of TAVI in patients with BAV is scarce, which might be due to the fact that TAVI practice in BAV is still not recommended by the guidelines.

Initially, with the implantation of the first generation of TAVI devices, the results were less convincing to the TAVI operators about the safety, in addition to the absence of a full understanding of the BAV anatomy in relation to the new THVs. Mylotte et al. were one of the first reports with one-year outcomes of TAVI in stenotic BAV in patients with moderate surgical risk, using two of the first-generation TAVI devices, CoreValve SEV (Medtronic, Inc., Minneapolis, MN, USA), and SapienXT BEV (Edwards Lifesciences, Inc., Irvine, CA, USA) [21], with all-cause mortality of 20.8% with SapienXT and 17.5% with CoreValve.

Multiple factors might affect the clinical and haemodynamic outcomes, such as the severity of the calcification [14,22], the residual AR, and the transprosthetic gradient.

With the significant calcifications associated with BAV anatomy, the risk of valve mal expansion is usually a concern, which may lead to significant paravalvular leakage, abnormal bioprosthetic leaflet geometry, and increased leaflet strain and stress with probable risk of early device failure and undesired effect on the valve durability [6,23,24].

Yoon et al. reported interesting findings regarding the association between the BAV calcification or raphe type and the outcomes in terms of all-cause mortality [14], with twofold increased risk at 1 year and fourfold at 2 years in patients with both calcified raphe and excess leaflets calcifications in comparison to patients without calcifications or raphe. All-cause mortality, irrespective of the calcification or raphe type, at 1 and 2 years was 6.7% and 12.5%, respectively, with cardiovascular mortality at 3.9% and 5.9%, respectively. In Yoon’s report, BAV patients were treated with TAVI using multiple devices, the majority (71.6%) with Sapien3 (Edwards Lifesciences, Inc., Irvine, CA, USA), 18.2% with Evolut R/Pro (Medtronic, Inc., Minneapolis, MN, USA), in addition to Lotus/edge, Accurate (Boston Scientific, Marlborough, MA, USA), and Portico (Abbott Structural Heart, Minneapolis, MN, USA) THVs. All-stroke was 2.7% compared to 3.2 observed in our report; pacemaker implantation was 12.2% vs. 8.3% in our report [14].

The 30-day report for the safety and efficacy of Myval THV in the same cohort showed encouraging results in terms of technical success, all-cause mortality, stroke, and residual AR [10]. This follow-up report, with a median duration of 13.6 months with the longest follow-up duration of up to 36 months in patients with low surgical risk, shows no significant change regarding the different haemodynamic parameters, transprosthetic gradient, EOA, and residual AR, which might suggest a stable performance of the device over the reported follow-up period (Figure 1).

One of the main questions regarding TAVI in BAV is: do the suspected outcomes outweigh the risk of THV failure? Especially in patients with expected longer life expectancy, and how the outcomes will impact the quality of life and the possibility of the need for valve reintervention in the future [3].

Now the evidence of TAVI safety and efficacy in TAV anatomy is strong enough [1,2], so one of the suggested study designs is the comparison of TAVI in BAV to TAV. Multiple retrospective studies with and without propensity score matching have been published in a trial to propose such a study and to present more trustable results [11,25].

Makkar et al. [13] conducted a retrospective study which included a propensity score matching between BAV and TAV in patients with low surgical risk and treated with TAVI using BEV Sapien3. Interestingly there was no significant difference between BAV and TAV in terms of all-cause mortality at 30-day (0.9% vs. 0.8%; HR, 1.18 [95% CI, 0.68 to 2.03]; *p* = 0.55) and 1 year (4.6% vs. 6.6%; HR, 0.75 [95% CI, 0.55 to 1.02]; *p* = 0.06), and stroke at 30-day (1.4% vs. 1.2%; HR, 1.14 [95% CI, 0.73 to 1.78]; *p* = 0.55) or 1 year (2.0% vs. 2.1%; HR, 1.03 [95% CI, 0.69 to 1.53]; *p* = 0.89). The authors have reported up to 1 year, the haemodynamic outcomes among BAV, treated with the 3rd generation Sapien3 and Sapien3-Ultra [13]. In comparison to our report, at 1-year, the mean PG was 13.2 vs. 10 mmHg, moderate or more PVL 3.4% vs. 2%.

The low rate of significant post-TAVI AR in this registry is similar to the Myval performance in patients with severe AS and tricuspid AV anatomy [26,27,28].

The recommendation and optimisation of TAVI in BAV is still an unmet need, and without a dedicated RCT comparing BAV and TAV, the evidence is still not strong enough.

In a recent report by Zhou et al., who reported up to 3 years, the outcomes in 109 BAV with low surgical risk in comparison to TAV in Chinese patients treated with multiple devices designs SEVs (83%), MEVs (9%), and BEVs (8%) [8]. All-cause mortality at 1-year, 2-year, and 3-year was 6.4%, 10.1%, and 12.8%, respectively. All-stroke was 4.6% at 2-year and 4.4 at 3-year. Regarding the hemodynamic outcomes, the Myval showed comparable results in comparison to Zhou et al. report at 1-year; despite the fact of SEVs were used in the majority of cases, the mean pressure gradient at 1-year was 11.4 mmHg vs. 10 mmHg, EOA 1.56 vs. 1.70 cm^2^, and moderate and severe PVL 8.3% vs. 2%. The pacemaker rate was higher than our report, 11.9% vs. 8.3%.

Despite TAVI being still not recommended in the BAV disease [1,2], multiple reports showed lower all-cause mortality when compared to TAV [8,13], maybe due to the facts of lower age, low surgical risk, in addition to patient selection.

The risk of early or late bioprosthetic valve dysfunction (BVD), including structural and non-structural dysfunction of TAVI devices with subclinical or clinical valve thrombosis and haemodynamic, is an important issue, without enough data reported yet [29]. This risk is linked mainly to difficult and heterogenous anatomy within BAV disease, which might cause eccentric and/or non-uniform device expansion with increased leaflet stress. The true incidence of valve leaflet thrombosis (LT) after TAVI in either BAV or TAV is not quite accurate, but still a concern and a hot research point. The classification of LT into clinical (symptomatic) and subclinical (asymptomatic) is one of the causes of this missing accurate incidence, as the clinical practice differs between countries and geographic areas, from the screening of most patients treated with TAVI for LT using MDCT (USA) to the limited indication to symptomatic patients or as a part from a specific study protocols (Europe and most of the other countries) [30]. The overall reported incidence of LT was 5.4%. The clinical LT was reported with less frequency of 1.2% in comparison to the sub-clinical LT of 15.1% [31]. In this study, there was no consensus on the screening of LT in all patients, and the MDCT scan was up to the investigator’s discretion and the clinical indication.

Accordingly, only one patient was diagnosed with clinical valve thrombosis, due to discontinuation of the antithrombotic therapy. After confirmation of the diagnosis by MSCT scan, the patient received thrombolytic therapy followed by improvement of the patient’s symptoms and haemodynamic assessment. In comparison to the 30-day echocardiographic assessment, 3 (6.4%) patients were diagnosed with stage 2 (moderate) haemodynamic deterioration, but they were asymptomatic. Two of them were associated with increased transprosthetic gradient and one due to an increase of AR by one grade to moderate AR. Patient with stage 3 (severe) haemodynamic deterioration was associated with a final transprosthetic gradient ≥30 mmHg. Until the collection of follow-up data for this report, no further investigations to identify the definite cause were performed or a decision on valve-related intervention.

Optimisation of TAVI in BAV has become an urgent unmet need, especially with the wide expansion of TAVI into younger and lower surgical populations in whom BAV is common [4,5].

The optimisation of the TAVI procedure should include detailed morphological characterisation (leaflets, raphe, and calcium distribution) and device selection according to the different anatomical features to provide the best outcomes regarding all clinical outcomes, hemodynamic performance, and coronary access patency. In addition, with such a young population planning for any future need for intervention, either valve-in-valve or coronary intervention must be in mind during procedure planning [6].

The obstacle facing the optimisation of TAVI in BAV might be due to the absence of RCT (s), which compares either TAVI vs. SAVR and others comparing the different THV devices [3,6].

The findings of this report as an initial assessment of the 1-year outcomes of TAVI using the Myval THV in BAV anatomy needs to be confirmed by results obtained from randomised trials, which may be confirmed by results from LANDMARK trial (NCT04275726) in which BAV is not excluded, and patients are randomised to either Myval, Sapien3 or Evolut Pro THV. In addition, COMPARE TAVI cohort-B trial (NCT04443023) does not exclude BAV, and patients are randomised to either Myval or Sapien3 THVs.

## 5. Limitations

Limitations of this registry include the retrospective nature of the registry. In addition, detailed anatomical characteristics (annular area and dimensions, calcium volume and distribution) were not collected systematically. The relatively small number of included patients, in addition to the differences in patient characteristics, when compared to the results of other studies with a large population number, is also a relevant limitation.

The absence of independent Corelab adjudication of the echocardiographic assessment is another limitation, but the participating sites have confirmed the follow-up of the recommended recent guidelines for the assessment of prosthetic valves by echocardiography. However, the results of initial experience using Myval in BAV are encouraging for safety and performance among different geographic regions.

## 6. Conclusions

The balloon expandable Myval THV implantation in patients with severe bicuspid AS shows favourable one-year hemodynamic and clinical outcomes, but these findings should be confirmed in a well-designed, adequately powered, and randomised study.

## Figures and Tables

**Figure 1 jcm-12-02398-f001:**
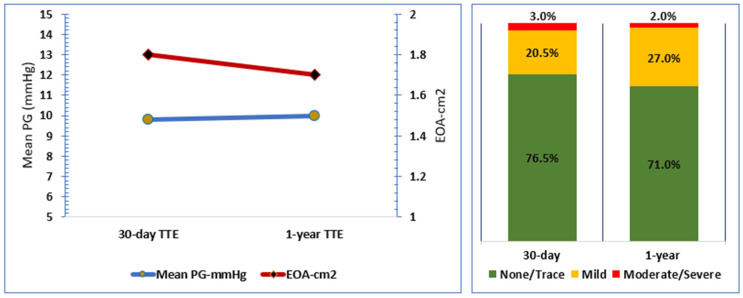
Haemodynamic performance at 30-day and 1-year follow-up.

**Table 1 jcm-12-02398-t001:** Patients’ baseline characteristics.

Demographic Characteristics	
Age	732 [66.3, 77.0]
Men	45 (72.6%)
Women	17 (27.4%)
Body surface area (BSA) m^2^	1.8 ± 0.27
Body mass index (BMI) kg/m^2^	25.3 ± 5.6
Clinical characteristics	
STS risk score%	3.2 ± 2.2
New York Heart Association (NYHA) class III/IV	40 (64.5)
Prior atrial fibrillation	10 (16.1%)
Peripheral vascular disease	8 (12.9%)
Bicuspid aortic valve (BAV) phenotype	
Type 0	10 (3.2%)
Type 1-a	50 (80.6%)
Type 2	2 (16.1%)

Data are presented as mean ± SD, median [IQR], or number (%).

**Table 2 jcm-12-02398-t002:** Clinical outcomes during follow-up.

Outcome	
Follow-up duration, months	13.5 [12.2, 18.3]
All-cause mortality	7 (11.3%)
Cardiovascular mortality	3 (4.8%)
Non-cardiovascular mortality	4 (6.5%)
TAVI-related rehospitalisation	1 (1.8%)
Other cardiovascular rehospitalisation	5 (8.8%)
Non-cardiovascular rehospitalisation	6 (10.5%)
All-stroke	2 (3.2%)
Myocardial infarction	1 (1.6%)
Permanent pacemaker implantation	5 (8.3%)
Major bleeding	1 (1.6%)
Acute kidney injury (AKI)	0
Endocarditis	0
Valve thrombosis	1 (1.6%)
Re-intervention to the valve	0
New York Heart Association (NYHA) Class	
NYHA I	27 (49.1%)
NYHA II	25 (45.5%)
NYHA III	3 (5.5%)
SARS COVID-2 infection	5 (8.2%)

Data are presented as median [IQR] or number (%).

**Table 3 jcm-12-02398-t003:** Echocardiographic (haemodynamic) outcome at one-year follow-up.

Echocardiographic Assessment	
Mean pressure gradient (mPG), mmHg	10 [8, 16.5]
Effective orifice area (EOA), cm^2^	1.7 [1.4, 1.9]
Transvalvular maximum velocity (Vmax), m/sec	2.1 [1.6, 2.6]
Left ventricle ejection fraction (EF), %	60 [55, 60]
Total aortic regurgitation (AR)	
None/Trace	37 (71%)
Mild	14 (27%)
Moderate/Severe	1 (2%)
Moderate/Severe mitral regurgitation	1 (2.3%)
Moderate/Severe tricuspid regurgitation	4 (9.5%)
Systolic pulmonary artery pressure (SPAP), mmHg	26 [23, 32]
Haemodynamic valve deterioration	
Stage II haemodynamic deterioration	3 (6.4%)
Stage III haemodynamic deterioration	1 (2.1%)

Data are presented as median [IQR] or number (%).

## Data Availability

Data are contained within the article.

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
