# Peer review of "One-Year Outcomes after Myval Implantation in Patients with Bicuspid Aortic Valve Stenosis—A Multicentre Real-World Experience"

_jcm, 2023, doi:10.3390/jcm12062398_

Round 1
Reviewer 1 Report
With interest, I read this report on mid-term outcomes of patients treated with a MyVal for severe bicuspid aortic valve stenosis. Here are my suggestions:
We need more information on follow-up. How many patients returned for a follow-up visit and when? How was the information retrieved in patients who did not return for a visit? How many patients have “mid-term” echo and how many paired echos are available? When were the “mid-term” echos performed? How many multidetector CT scans have been performed and when? Results must be interpreted in the light of completeness of the respective examinations. E.g. do we know the true frequency of asymptomatic valve thrombosis and thus of stage 1haemodynamic valve deterioration?
What were the causes of stage 2 and 3 haemodynamic valve deterioration? This information should be given in the results section.
When comparing outcomes between different registries, differences in baseline risk, i.e. STS-PROM, must be taken into account.
Please, expand on the frequency of haemodynamic valve deterioration in the discussion.
Line 60: „Cardiovascular” should read „cardiovascular“.
Author Response
Thank you very much for your valuable comments and suggestions, which we believe will help us improve the submitted article.
# Reviewer:
more information on follow-up.
- How many patients returned for a follow-up visit and when?
- How was the information retrieved from patients who did not return for a visit?
#Reply:
- Fifty-five patients returned for a follow-up visit with a median of follow-up duration of 13.5 [12.2, 18.3] months from the index TAVI procedure (lines 148, 149).
- In the patients and methods section (Lines 114-116) the authors mentioned how the follow-up of clinical outcomes data have been retrieved “Patients follow-up was performed according to each site local protocol (in-person visits or structured telephone calls). And in case of the site didn’t reach the patient and patients who didn’t return for the planned follow-up or were unable to be followed by phone were excluded from the study, this is why this report included only 62 patients in comparison to the early 30-day report which included 68 patients. So, six patients were not included in this report.
# Reviewer:
- How many patients have “mid-term” echo and how many paired echos are available?
- When were the “mid-term” echo performed?
#Reply:
- Fifty (50) patients had echo studies at the mid-term follow-up with 47 patients with paired Echo studies at the 30-day and midterm follow-up.
- The mid-term echo performed at 1 year or further (in this report up to 36 months).
- Due to the lack of consensus about the exact definition of mid-term time point, and most of the patients’ follow-up in this article was within 18 months from the index procedure, we found that changing the title to “One-year Outcomes after Myval Implantation in Patients with Bicuspid Aortic Valve Stenosis – A Multicentre Real-World Experience” will be more convincing and to avoid misleading title.
#Reviewer:
- How many multidetector CT scans have been performed and when?
- Results must be interpreted in the light of the completeness of the respective examinations. E.g. do we know the true frequency of asymptomatic valve thrombosis and thus of stage 1haemodynamic valve deterioration?
- What were the causes of stage 2 and 3 haemodynamic valve deterioration? This information should be given in the results section.
#Reply:
- MDCT scan was performed for only one patient who was diagnosed with witnessed valve thrombosis, due to discontinuation of the recommended antithrombotic strategy, after that the patients presented with acute and worsening symptoms, so the MDCT was mandatory in such a situation. The acute presentation and scan were done 12 months after the index TAVI procedure. The MDCT for the diagnosis of valve thrombosis was up to the investigator’s discretion and the clinical indications.
- Regarding this point we agree with the reviewer’s valuable comment, but as you know in Europe and other countries (like India) and with the exception of the USA, routine post-TAVI MDCT scans for screening of asymptomatic valve thrombosis (HALT, HAM, and RELM) is not the usual practice and only can be done with an indication like what occurred in our patients or as a part of specific protocol. So, the true incidence of asymptomatic valve thrombosis will still be a tough question to be answered apart from the limitations of the difference in practice between the different geographic areas.
- Regarding the other patients with either stage II (three patients) or III (one patient) haemodynamic deterioration all of them were asymptomatic so they didn’t undergo MDCT scan till the last follow-up ( mentioned in lines 273-279).
# Reviewer:
- When comparing outcomes between different registries, differences in baseline risk, i.e. STS-PROM, must be taken into account.
#Reply:
- Thank you very much for this important point, we tried to mention the essential patient characteristics for each study we used to compare with this study. Like the surgical risk,
- It was missing with Mylotte et al report and now it is added.
#Reviewer:
- Please, expand on the frequency of haemodynamic valve deterioration in the discussion.
#Reply:
- The paragraph has been expanded to include that the true incidence of asymptomatic valve thrombosis is still a concern and a hot research topic. In addition to the study methodology regarding the diagnosis of valve thrombosis.
#Reviewer:
- Line 60: „Cardiovascular” should read „cardiovascular“.
#Reply:
- Corrected

Reviewer 2 Report
This study reports the unique experience with a new THV in bicuspid aortic valve. Bicuspid valve is one of the last remaining challenges of TAVR and it's important to evaluate new devices in this anatomy.
Below are few comments aimed to improve the content of the paper.
A comparison group would be meaningful to understand the effectiveness of this device in bicuspid patients. Could the authors provide data of tricuspid patients implanted with Myval prosthesis during the same time period ?
1 year mean follow-up is not mid-term but short-term follow-up. Please change it throughout the manuscript as it could be misleading for the reader.
One major limitations is the absence of morphological data. Authors claimed that it was not collected systematically but nowadays, TAVR is never performed without CT-scan analysis. It should be possible to get those data :
-Could the authors provide the aortopathy phenotype in baseline characteristics, root and aorta dimensions and aorta-valve angle. Annulus dimensions would be appreciated too.
-The two main prognostic parameters in bicuspid patients are excess leaflet calcification and calcified raphe (see Yoon and al, JACC 2020). Could the authors provide those data including calcium score and volume in baseline CT characteristics.
Please add "1-year" in the title of Table 3 and Figure 1
Author Response
Dear Reviewer, thank you very much for your valuable comments and suggestions, which we believe will help to improve the content of the submitted article.
#Reviewer:
This study reports the unique experience with a new THV in bicuspid aortic valve. Bicuspid valve is one of the last remaining challenges of TAVR and it's important to evaluate new devices in this anatomy.
Below are a few comments aimed to improve the content of the paper.
A comparison group would be meaningful to understand the effectiveness of this device in bicuspid patients. Could the authors provide data of tricuspid patients implanted with Myval prostheses during the same time period?
#Reply:
Thank you very much for this important point, with the fact of most the published data about the effectiveness and safety of Myval THV in patients with severe AS are obtained from non-randomised studies, and the large randomised studies including Myval THV series are still running (LANDMARK trial (NCT04275726) and COMPARE TAVI cohort-B trial (NCT04443023)), this is considered one of the retrospectives, observational studies which report the initial data of Myval usage in stenotic BAV.
The nature of this study is to include only patients with BAV as there are other many studies which reported the initial results in tricuspid AV, we cited the most important studies with patients with tricuspid AV for the comparison of some outcomes like residual AR (lines 250, 251).
#Reviewer:
1 year mean follow-up is not mid-term but short-term follow-up. Please change it throughout the manuscript as it could be misleading for the reader.
#Reply
Thank you very much for your comment regarding the used term “mid-term follow up”, it looks like there is no consensus on the exact definition of short-term, mid-term and long-term time points and sometimes as mentioned may be confusing or misleading.
We used the mid-term outcome as around 75% of patients included patients in this study had a follow-up duration of more than 12 months and up to 36 months. So the median follow-up duration was 13.5 [12.2, 18.3].
Based on the definitions proposed by other studies and the VARC3, we considered the early outcomes as 30-day (as reported in the 1st report for the same cohort (https://doi.org/10.3390/jcm11020443), and short term below one year, So to avoid the misleading and confusion and due to most of the patients’ follow-up we within 18 months with median follow up of 13.5 [12.2, 18.3] months from the index TAVI procedure the title has been changed to One-year Outcomes after Myval Implantation in Patients with Bicuspid Aortic Valve Stenosis – A Multicentre Real-World Experience”.
#Reviewer:
One major limitation is the absence of morphological data. Authors claimed that it was not collected systematically but nowadays, TAVR is never performed without CT-scan analysis. It should be possible to get those data.
#Reply:
Thank you very much for this comment, as mentioned this is one of the limitations of this study (mentioned in the study limitations), we confirm that all patients were treated with TAVI according to the sizing and preprocedural analysis of MDCT scans as recommended, but we were unable to collect all the required data of the MDCT analysis due to some the logistic difficulties in some centres, this what was referred to as “not collected systematically” so the frequency of the collected data was below the accepted range to be reported, so we only reported the BAV phenotype which reported in all patients.
#Reviewer:
Could the authors provide the aortopathy phenotype in baseline characteristics, root and aorta dimensions and aorta-valve angle? Annulus dimensions would be appreciated too.
#Reply:
These are also some of the non-collected data as mentioned previously, but the investigators confirmed the absence of a clear indication for the surgical intervention for the aortic root or ascending aorta and the local heart team’s decision to treat those young patients with low surgical risk with TAVI was supported by that.
#Reviewer:
The two main prognostic parameters in bicuspid patients are excess leaflet calcification and calcified raphe (see Yoon and al, JACC 2020). Could the authors provide those data including calcium score and volume in baseline CT characteristics?
#Reply:
Thank you very much for your comment and suggestion regarding this point. We already cited Yoon and al, JACC 2020, as it is one of the important reports about the outcomes in AS with BAV after treatment with TAVI.
Due to the retrospective nature of the data, the use of site-reported data, in addition to the fact of the detailed calcium analysis is not routinely done for all TAVI cases we were not able to collect such data, and just reported the outcomes including the mortality without specific analysis of its link with any specific another variable.
#Reviewer:
Please add "1-year" in the title of Table 3 and Figure 1
#Reply:
Mid-term follow-up was added.

Reviewer 3 Report
BRIEF SUMMARY OF ARTICLE
The reviewed research article presented a retrospectively conducted multi-center study comprising mid-term (median 13.5 years) follow-up results of 62 patients who underwent TAVI with the Myval balloon-expandable valve for symptomatic severe stenosis of bicuspid aortic valve morphology.
Clinical outcome parameters were oriented at the VARC-3 recommendations. All-cause mortality occurred in 7 (11.3%), rehospitalization in 12 (21.1%), stroke in 2 (3.2%), and permanent pacemaker implantation in 5 (8.3%) patients. As a hemodynamic result, mean transprosthetic pressure gradient was reported as 10 [8, 16.5] mmHg, and ≥moderate AR was seen in 1 (2%) patient.
GENERAL COMMENTS
The manuscript is well-structured and focuses on an interesting as well as clinically-relevant topic. The assessment and results TAVI with Myval in bicuspid aortic valve morphology and severe symptomatic stenosis are adequately presented in the text and sufficiently supported by multiple tables&figures. The statistical tests applied here seem ok. So are the conclusions drawn from the results – although these should be alleviated to some degree, because of the limitations that apply to this study. Overall, the methods of the contributed submitted manuscript follow a reasonable structure that is maintained throughout the manuscript, although there are some methodological limitations that are to be addressed or at least sufficiently pointed out.
SPECIFIC COMMENTS
Title:
I would recommend to exclude the name of the novel balloon-expandable valve from the title to reduce marketing overtones. Instead, I would keep it more descriptive e.g. “novel balloon-expandable transcatheter heart valve”.
Abstract:
The abstract uses the name and brand name of the THV which I would recommend to avoid in the abstract, but describe it mainly in the methods section, please see above.
Introduction:
- Throughout the manuscript there are linguistic and grammatical limitations (e.g. line 75 “Bicuspid” should not be written with capital letter; in line 90 it should read “are scarce” instead of “is scarce”; there is also a superfluent blank space in line 90 after the first prase, etc. etc.). Therefore, a final native language revision should be performed.
- Instead of the Sievers’ classification of bicuspid aortic valve morphology alone, the authors should include and consider more recent and relevant classification of Yoon SH et al. (Bicuspid Aortic Valve Morphology and Outcomes After Transcatheter Aortic Valve Replacement. J Am Coll Cardiol. 2020 Sep 1;76(9):1018-1030. doi: 10.1016/j.jacc.2020.07.005)
- The length of the introduction is ok and concise.
Materials and Methods:
- the authors should classify bicuspid aortic valve morphology according to the recommendation of Yoon et al., please see above.
- was screening for subclinical valve thrombosis performed? How was the strategy of diagnosing valve thrombosis in the cohort?
- more detailed description of statistics should be used, normality test, etc.?
Discussion
- when comparing the results to relevant previous literature, authors should attempt to explain differences, instead of only listing the results, e.g. why was all-cause mortality less than shown by Mylotte et al., why was the permanent pacemaker rate lower than in Yoon et al., etc!
- The main results and conclusion should be moderated, because of the limitations that apply to the study (small n of cohort; retrospective; heterogeneity of bicuspid anatomy)
References:
Ok
Tables/figures:
Ok
Abbreviations:
- Abbreviations should be more stringently used, e.g. “LVOT” ist not listed, although used in manuscript; “PMI” is listed as an abbreviation but never used; “bioprosthetic valve dysfunction (BVD)” is introduced and used only once in the text, therefore is no point in introducing this abbreviation.
Author Response
Thank you very much to the reviewer for all the valuable comments, we believe that will help to improve the submitted article.
#Reviewer:
GENERAL COMMENTS
The manuscript is well-structured and focuses on an interesting as well as clinically-relevant topic.
The assessment and results of TAVI with Myval in bicuspid aortic valve morphology and severe symptomatic stenosis are adequately presented in the text and sufficiently supported by multiple tables & figures.
The statistical tests applied here seem ok.
So are the conclusions drawn from the results – although these should be alleviated to some degree, because of the limitations that apply to this study.
Overall, the methods of the contributed submitted manuscript follow a reasonable structure that is maintained throughout the manuscript, although there are some methodological limitations that are to be addressed or at least sufficiently pointed out.
#Reply:
Thank you very much for your positive comments.
Regarding the conclusion, we added “but these findings should be confirmed in well-designed, adequately powered and randomized studies.”
Some of the methodological limitations have been added to the limitation section.
#Reviewer:
SPECIFIC COMMENTS
# Reviewer:
Title:
I would recommend excluding the name of the novel balloon-expandable valve from the title to reduce marketing overtones. Instead, I would keep it more descriptive e.g. “novel balloon-expandable transcatheter heart valve”.
#Reply:
Thank you very much for your comment, regarding this point we realize that the aim is to avoid the commercial sounds with such scientific work, but the in the first report title it was mentioned as “Myval”, in addition, the term “ the novel balloon expandable” might be no longer acceptable to be used with this generation of Myval as it is now more than three years since the 1st use of the BEV Myval, in addition to the introduction of other new balloon-expandable devices technologies in the market, one of them is the 2nd iteration of the Myval which will be introduced in Europe soon.
Also, we would like to address that the title has been changed to “One-year Outcomes after Myval Implantation in Patients with Bicuspid Aortic Valve Stenosis – A Multicentre Real-World Experience”.
Due to the lack of consensus about the exact definition of Mid-term follow-up and to avoid any misleading
#Reviewer
Abstract:
The abstract uses the name and brand name of the THV which I would recommend avoiding in the abstract, but describe it mainly in the methods section, please see above.
#Reply:
As previously mentioned, as it is mentioned in the title of the previous and current report, we found that it might be acceptable to be clearly mentioned and to avoid any confusion with any other balloon expandable devices in the market.
#Reviewer:
Introduction:
- Throughout the manuscript, there are linguistic and grammatical limitations (e.g. line 75 “Bicuspid” should not be written with a capital letter; in line 90 it should read “are scarce” instead of “is scarce”; there is also a superfluent blank space in line 90 after the first prase, etc. etc.). Therefore, a final native language revision should be performed.
- Instead of Sievers’ classification of bicuspid aortic valve morphology alone, the authors should include and consider a more recent and relevant classification of Yoon SH et al. (Bicuspid Aortic Valve Morphology and Outcomes After Transcatheter Aortic Valve Replacement. J Am Coll Cardiol. 2020 Sep 1;76(9):1018-1030. doi: 10.1016/j.jacc.2020.07.005)
- The length of the introduction is ok and concise.
#Reply:
Thank you very much for these comments, some of the linguistic errors have been corrected in addition we will consider a language revision for the all article.
We decided to use the famous and well-known Sievers’ classification to facilitate the reporting of the BAV phenotypes, Yoon et al, used only Raphe (for type 1 BAV) and non-raphe (for type 0 BAV), but they didn’t include or describe two raphe (type 2 BAV) and this cohort include two patients with type 2 BAV.
Also, we considered the used classification in Yoon et al report was mainly used to report the outcomes in relation to specific and sophisticated analysis of the AV phenotype in terms of the presence and absence of raphe (BAV type 0 and type 1 only) and the degree of calcification within the raphe, and this is not the case in our study.
#Reviewrs:
Materials and Methods:
- the authors should classify bicuspid aortic valve morphology according to the recommendation of Yoon et al., please see above.
- was screening for subclinical valve thrombosis performed? How was the strategy of diagnosing valve thrombosis in the cohort?
- more detailed description of statistics should be used, normality test, etc.?
#Reply:
Regarding the use of the Yoon et al classification, it is mentioned in the reply to the previous comment.
There was no routine screening for valve leaflet thrombosis as a part of this study, it was up to the investigator’s discretion and the clinical indications, as the practice in most of Europe and geographic regions other than the USA is to order the MDCT in post-TAVI situation if there is a clinical indication or as a part of a specific trial after getting patients consent and ethical approval.
Regarding the statistical tests, the Shapiro-wilk normality test was mentioned as conducted firstly to determine the distribution of the data (line 134).
#Reviewer:
Discussion
- when comparing the results to relevant previous literature, authors should attempt to explain differences, instead of only listing the results, e.g. why was all-cause mortality less than shown by Mylotte et al., why was the permanent pacemaker rate lower than in Yoon et al., etc!
- The main results and conclusion should be moderated, because of the limitations that apply to the study (small n of the cohort; retrospective; heterogeneity of bicuspid anatomy).
#Reply:
Thank you for your comments regarding these points,
For the Mylotte et al study we clearly stated that “Mylotte et al., was one of the first reports with 1-year outcomes of TAVI in stenotic BAV in patients with moderate surgical risk, using two of the first generation TAVI devices, CoreValve SEV (Medtronic, Inc., Minneapolis, Minnesota) and SapienXT BEV (Edwards Lifesciences, Inc., Irvine, California), with all-cause mortality of 20.8% with SapienXT and 17.5% with CoreValve.” And the use of those early THV generations with frequent undesirable outcomes might be contributed to such high mortality (lines 199-206)
For the comparison with Yoon et al study, we mentioned that different TAVI devices were used, including some devices with known high rates of complications like Evolut and Lotus which were associated with high pacemaker implantation, ACURATE and Portico were associated with more significant PVL. And we mentioned all of the devices used in this study as a possible explanation for the difference in outcomes.
Regarding the conclusion and to make it moderate as mentioned we added the following sentence “but these findings should to be confirmed in a well-designed, adequately powered and randomized studies.” And the limitations of the study are described in the limitations section.
#Reviewer:
Abbreviations:
- Abbreviations should be more stringently used, e.g. “LVOT” ist not listed, although used in manuscript; “PMI” is listed as an abbreviation but never used; “bioprosthetic valve dysfunction (BVD)” is introduced and used only once in the text, therefore is no point in introducing this abbreviation.
#Reply:
The abbreviation has been reviewed.
Words used less frequently (less than five times were eliminated from the abbreviation list).
LVOT was mentioned once, and BVD was mentioned once, so both were not included in the list.

Round 2
Reviewer 1 Report
With interest, I read this report on mid-term outcomes of patients treated with a MyVal for severe bicuspid aortic valve stenosis. Here are my suggestions:
We need more information on follow-up. How many patients returned for a follow-up visit and when? How was the information retrieved in patients who did not return for a visit? How many patients have “mid-term” echo and how many paired echos are available? When were the “mid-term” echos performed? How many multidetector CT scans have been performed and when? Results must be interpreted in the light of completeness of the respective examinations. E.g. do we know the true incidence of asymptomatic valve thrombosis and thus of stage 1 haemodynamic valve deterioration?
What were the causes of stage 2 and 3 haemodynamic valve deterioration? This information should be given in the results section.
When comparing outcomes between different registries, differences in baseline risk, i.e. STS-PROM, must be taken into account.
Line 60: „Cardiovascular” should read „cardiovascular“.
Author Response

(The authors gave the same response as above.)
